# Useful Clinical Criteria for Identifying Neonates with Congenital Cytomegalovirus Infection at Birth in the Context of an Expanded Targeted Screening Program

**DOI:** 10.3390/v16071138

**Published:** 2024-07-16

**Authors:** Valeria Poletti de Chaurand, Gaia Scandella, Marianna Zicoia, Francesca Arienti, Federica Fernicola, Laura Lanteri, Diletta Guglielmi, Anna Carli, Maria Viola Vasarri, Lucia Iozzi, Annalisa Cavallero, Sergio Maria Ivano Malandrin, Anna Locatelli, Maria Luisa Ventura, Mariateresa Sinelli, Sara Ornaghi

**Affiliations:** 1Obstetric Unit, Foundation IRCCS San Gerardo dei Tintori, 20900 Monza, MB, Italy; v.polettidechaura@campus.unimib.it (V.P.d.C.); g.scandella5@campus.unimib.it (G.S.); f.arienti9@campus.unimib.it (F.A.); f.fernicola@campus.unimib.it (F.F.); l.lanteri@campus.unimib.it (L.L.); d.guglielmi4@campus.unimib.it (D.G.); annacarli88@gmail.com (A.C.); m.vasarri@campus.unimib.it (M.V.V.); anna.locatelli@unimib.it (A.L.); 2School of Medicine and Surgery, University of Milan-Bicocca, Via Cadore 48, 20900 Monza, MB, Italy; m.zicoia@campus.unimib.it; 3Neonatal Intensive Care Unit, Foundation IRCCS San Gerardo dei Tintori, 20900 Monza, MB, Italy; lux_cis@hotmail.it (L.I.); mlois.ventura@gmail.com (M.L.V.); mariateresa.sinelli@irccs-sangerardo.it (M.S.); 4Microbiology Unit, Foundation IRCCS San Gerardo dei Tintori, 20900 Monza, MB, Italy; annalisa.cavallero@irccs-sangerardo.it (A.C.); sergiomaria.malandrin@irccs-sangerardo.it (S.M.I.M.)

**Keywords:** cytomegalovirus, screening, targeted, newborn, congenital, infection

## Abstract

Cytomegalovirus (CMV) is the leading infectious cause of brain defects and neurological dysfunctions, including sensorineural hearing loss (SNHL). Targeted screening in neonates failing the hearing screen is currently recommended in Italy according to national guidelines. However, SNHL may not be present at birth; also, congenital CMV (cCMV) may manifest with subtle signs other than SNHL. Therefore, the inclusion of additional criteria for cCMV screening appears clinically valuable. Starting January 2021, we have implemented expanded targeted cCMV screening at our center, with testing in case of maternal CMV infection during pregnancy, inadequate antenatal care, maternal HIV infection or immunosuppression, birthweight and/or head circumference < 10th centile, failed hearing screen, and prematurity. During the first three years of use of this program (2021–2023), 940 (12.3%) of 7651 live-born infants were tested. The most common indication was birthweight < 10th centile (n = 633, 67.3%). Eleven neonates were diagnosed as congenitally infected, for a prevalence of 1.17% (95%CI 0.48–1.86) on tested neonates and of 0.14% (95%CI 0.06–0.23) on live-born infants. None of the cCMV-infected newborns had a failed hearing screen as a testing indication. Implementation of an expanded cCMV screening program appears feasible and of clinical value.

## 1. Introduction

Congenital cytomegalovirus (cCMV), with a prevalence of approximately 0.2–2.2% [1], is the leading infectious cause of brain defects and neurological dysfunctions in newborns and children [2,3].

Approximately 10–15% of cCMV-infected infants show symptoms at birth, such as microcephaly, intracranial calcifications, sensorineural hearing loss (SNHL), petechiae, and retinitis [4].

In contrast, 85–90% of congenitally infected newborns have no such clear evidence of disease at birth, thus making diagnosis challenging in the absence of a newborn screening program. Clinical presentation can be non-specific and subtle, including low birthweight (BW), head circumference (HC) < 10th centile, jaundice, hepatosplenomegaly, and mild thrombocytopenia [5,6]. Importantly, 10–15% of the congenitally infected, asymptomatic neonates will develop late-onset symptoms of cCMV infection, among which SNHL is the most common [7,8]. In the absence of screening, these babies do not receive a diagnosis and, therefore, an appropriate follow-up, and almost none of them is identified until speech and language delays are obvious [5].

Although several studies have suggested the potential benefits of universal newborn screening for cCMV to detect all infants at risk of sequelae, there is still an ongoing debate about the feasibility of such a program [9,10,11,12,13,14,15,16].

Italian guidelines do not recommend universal newborn cCMV screening [17]. However, considering that SNHL is the most common cCMV-related sign in otherwise apparently healthy neonates and cCMV is the most common non-genetic cause of childhood SNHL [18], these guidelines highlight the importance of performing cCMV screening in all neonates failing the hearing screen, which is indeed universally recommended in Italy [17,19,20]. Unfortunately, this targeted cCMV screening program fails to identify those infected neonates without SNHL at birth [21,22,23].

In this context, it is important to identify other potential subtle clinical manifestations of cCMV, such as low BW, that could appropriately indicate the need for newborn screening, thus facilitating a timely diagnosis [6,24].

Findings from the implementation and routine clinical use of such expanded targeted programs in centers located in North America, Japan, and Israel have been recently reported, highlighting their feasibility and improved detection rates of cCMV-infected neonates compared to cCMV testing under a hearing-targeted-only approach [14,25,26,27,28,29]. In contrast, no data are available regarding expanded targeted cCMV newborn screening programs in Italy.

Here we present the results of the first three years of use of such a screening program at our academic maternity center, located in Northern Italy, and detail the main indications prompting cCMV screening as well as incidence of symptomatic and asymptomatic cCMV at birth in our study population.

## 2. Materials and Methods

### 2.1. Study Population

This is a prospective observational study including all neonates born at our university maternity center and screened for cCMV infection within 21 days after birth, between January 2021 and December 2023.

Since 2018, a hearing-targeted cCMV screening program has been active at our institution, with testing performed in neonates failing the hearing screen. Starting 1 January 2021, this screening program was expanded, with cCMV testing performed on live-born neonates if one of the following conditions was present: suspected or confirmed maternal CMV infection during pregnancy, antenatal diagnosis of fetal growth restriction (FGR) according to Delphi criteria [30], antenatal ultrasound suspicion of fetal infection (e.g., hyperechoic bowel, liver anomalies, ventriculomegaly, etc.), antenatal amniocentesis positive for CMV DNA, maternal HIV infection with detectable viral load and/or suppressed CD4+ count, maternal immunosuppressive therapy, prematurity < 37 weeks’ gestation, BW and/or HC < 10th centile according to InterGrowth-21 (IG-21) charts [31], failed neonatal hearing screening, thrombocytopenia (platelet count < 150 platelets/microL), or other neonatological indications (e.g., hepatomegaly, hepatitis). In January 2023, an additional criterion for testing was added: inadequate antenatal care (i.e., first antenatal visit after 14 weeks’ gestation or less than 3 antenatal assessments throughout gestation).

In order to assess all newborns with indications for cCMV infection screening according to our expanded protocol, we simultaneously implemented a standardized medical record system which monitors requests for CMV PCR testing, including the reasons for testing. Also, whenever a condition requiring cCMV testing is typed into the system, an alert is generated to prompt the assisting physician into ordering it.

For all newborns tested, we collected the mother’s medical data, course of pregnancy, gestational age at delivery, and delivery mode, in addition to the newborn’s day of birth, BW and BW centile, HC centile, APGAR score at the 1st and 5th minute of life, cord blood pH value, and the results of blood tests, s physical examination, and instrumental and hearing tests carried out in the early days of life. Whenever available, maternal serological status for CMV was also collected. Of note, the 2011 Italian guideline on low-risk pregnancy recommended against universal serological screening for CMV in pregnancy [32], thus resulting in an unknown maternal serological status in some of our newborns. At our center, maternal CMV IgG, IgM, and IgG avidity are measured by a chemiluminescence method on LIAISON^®^ (DiaSorin, Saluggia, Italy), following the manufacturer’s recommendations.

Both an antenatal and postnatal outpatient clinic for perinatal infections is available at our institution for women with suspected or confirmed CMV infection in pregnancy or fetal CMV infection and their newborns, respectively. For those pregnant women undergoing serological screening and being diagnosed with primary CMV infection during the peri-conceptional period or in pregnancy within 24 weeks’ gestation, valacyclovir (VCV) was administered (8 g/day) according to the Italian Drug Agency (AIFA) statement [33].

### 2.2. Sample Collection and CMV DNA Detection

In our study, screening was performed by CMV DNA identification on saliva samples [16]. In case of positivity, a urine sample was assessed for confirmatory purposes [15]. In some cases, particularly those born during the initial months after protocol implementation, testing was performed only on urine samples.

Saliva samples were collected using a cotton swab (COPAN^®^, Brescia, Italy) positioned in the cheek to collect pooled saliva (UTM-RT transport medium, COPAN^®^, Brescia, Italy). Collection would occur at least two hours after the last breastfeeding to limit false positive results. The samples were stored at 4 °C until they were transported to the hospital laboratory within 6 h. Urine samples were collected using an adhesive perineal bag for newborns (urine specimen collection bag, MedEvolution^®^, Changzhou DSB Medical Co., Ltd., Changzhou, China).

CMV testing was performed firstly with a full off board extraction by magnetic beads technology using InGenius instruments (ELITechGroup S.p.a., Turin, Italy). Then, through isothermal nucleic acid amplification by an ELITechGroup CMV ELITe MGB^®^ kit (ELITechGroup S.p.a., Turin, Italy), consisting in a real-time polymerase chain reaction (rt-PCR) method automatically conducted on the AB 7500 fast dx thermocycler (Applied Biosystems Italia, Monza, Italy). The results were elaborated by the software and expressed in a quantitative way (limit of detection: <650 copies/mL).

Patients were classified as cCMV-infected if they had a positive PCR result on urine. In these cases, viral load was also quantified on whole blood.

### 2.3. Management of cCMV-Infected Newborns

Once the positivity is found, patients undergo a standard workup, including complete blood count, hepatic and kidney serum profiles, hearing testing (otoacoustic emission—OAE, automatic auditory brainstem response—aABR, and ABR threshold exam), ophthalmologic visit with fundus oculi examination, head and abdomen ultrasound, neuropsychiatric visit, and brain magnetic resonance imaging (MRI) [34].

For all positive infants, the follow-up protocol includes ongoing clinical assessments, with evaluations by neuropsychiatrists and hearing assessments until the age of 6 years.

According to the 2017 Expert Consensus Statement by the European Society of Pediatric Infectious Disease (ESPID) on management of cCMV-infected newborns [35], symptomatic infants are treated within 30 days of life with Valgancyclovir (ValGCV) 16 mg/kg/dose every 12 h orally, substituted with intravenous Gancyclovir (GCV) 6 mg/kg/dose in cases of food intolerance. Antiviral treatment in neonates with isolated sensorineural hearing loss (SNHL) is discussed on an individual basis. ValGCV is continued for a duration ranging from 6 weeks to 6 months.

### 2.4. Statistical Analysis

Descriptive statistics were employed to describe the study population, with absolute and relative frequencies for categorical variables, and mean ± standard deviation or median and interquartile range for normally and not-normally distributed continuous variables, respectively. Distribution of continuous variables was assessed visually.

Overlapping indications for testing were considered separately for the calculation of prevalence by testing indication.

The prevalence rate of cCMV infection was calculated as the number of cCMV neonates per 100 live births with a 95% confidence interval (CI), assuming the Poisson approximation to the binomial distribution. Analyses have been performed for the overall prevalence of cCMV infection, to determine the detection rate of our expanded screening program, and for symptomatic and asymptomatic cCMV at birth.

SPSS (IBM SPSS Statistics for Macintosh, Version 28.0, IBM Corp, Armonk, NY, USA) and Prism GraphPad (version 10.0.0 for Macintosh, GraphPad Software, Boston, MA, USA) were employed for the analyses.

### 2.5. Ethical Considerations

This study was conducted in accordance with the Declaration of Helsinki and approved by the Brianza Ethics Committee (protocol code 3156, date of approval 30 January 2020). Informed consent on the use of anonymized data was obtained from the parents or legal guardians of all subjects involved in this study.

## 3. Results

During the study period, a total of 7651 infants were live-born, 940 (12.3%) of whom were tested for cCMV infection. One hundred and fourteen (12.1%) neonates were from multiple gestations, and in 36 of them, both twins had indications for cCMV testing. Table 1 displays general charateristics of the mothers and their newborns.

Most of the tested babies (n = 699, 74.4%) were born at term after 37 weeks’ gestation, at a mean gestational age of 38^2/7^ weeks (range 22^6/7^–42^1/7^ weeks).

The most common indication for cCMV testing was BW < 10th centile (n = 633, 67.3%), followed by HC < 10th centile (n = 270, 28.7%), and preterm birth (n = 241, 25.6%) (Figure 1). Maternal CMV infection indicated cCMV screening in 42 (4.5%) neonates, whereas 22 newborns were tested because of a failed hearing screen (2.3%). No cases had maternal HIV with detectable viral load and/or suppressed CD4+ count as indication for cCMV testing. All neonates were tested for cCMV within 21 days of life.

Of the 940 tested newborns, 66.6% (n = 626) met one criterion, 25.9% (n = 243) two criteria, and 7.6% (n = 71) three or more criteria for testing.

There were no cases with indication for cCMV infection screening who were not tested.

Saliva PCR was performed as a first screening test in almost all newborns (n = 917, 97.6%); in 23 (2.4%) cases, a saliva sample was not collected, and the screening test was performed on urine. No cases of false positive results with saliva PCR were registered.

Eleven newborns were diagnosed as congenitally infected, for a prevalence of 1.17% (95%CI 0.48–1.86) on tested neonates and of 0.14% (95%CI 0.06–0.23) on live-born infants (Table 2).

The highest prevalence of cCMV was among infants tested because of maternal CMV infection (8/42, 19%), followed by thrombocytopenia (4/67, 6%), preterm birth (3/241, 1.2%), HC < 10th centile (3/270, 1.1%), and BW < 10th centile (6/633, 0.95%). None of the eleven cCMV-infected newborns had a failed hearing test at birth, but two (cases n. 4 and 10) showed an abnormal result of the ABR threshold exam at two months and 3 weeks of age, respectively, which prompted ValGCV therapy initiation; normal findings were recognized at follow-up. In both cases, a maternal primary CMV infection was diagnosed.

In four cases (n. 4, 5, 9, and 11), the sole indication for neonatal testing was maternal CMV infection, either primary or non-primary, thus leading to a prevalence of asymptomatic cCMV infection at birth of 0.42% (95%CI 0.84–0.01). In the remaining seven cases, cCMV infection followed the ESPID criteria for being symptomatic at birth (prevalence 0.74%, 95%CI 0.20–1.29). Of note, case n. 3 had a deficient growth as the only indication for testing.

Maternal CMV infection, alongside antenatal ultrasound signs suggestive of fetal infection and an amniocentesis positive for CMV DNA, was an additional criterion for cCMV testing in cases n. 1 and 8; case n.1 also presented several other indications for testing at neonatal assessment after birth (Table 2).

Urine and whole blood were assessed in all eleven cCMV neonates: urine was CMV DNA-positive in all of them with a mean viral load of 55,879,733.09 copies/mL, whereas whole blood tests were positive in ten out of eleven cCMV-infected neonates, with a mean viral load of 2,278,113.90 copies/mL. Eight (72.7%) of the congenitally infected patients were saliva tested, with a mean CMV viral load of 20,276,203.25 copies/mL.

An adverse clinical outcome was observed in two of the congenitally infected babies (n. 1 and 2), whereas the remaining neonates showed either a mildly abnormal (cases n. 3 and 6) or a regular follow-up.

The clinical case of neonate n. 1 has been previously published [36]. Briefly, there was a diagnosis of fetal ascites and hyperechoic bowel at 20^4/7^ weeks, and the analysis of amniotic fluid revealed CMV DNA. The woman was started on VCV 8g/day according to the AIFA statement [33]. Fetal brain anomalies, confirmed by MRI, developed at 27 weeks, alongside with polyhydramnios at 33 weeks. At 34^1/7^ weeks an emergency cesarean section was performed for a non-reassuring fetal heart rate with the birth of a female neonate weighing 1550 g. Immediately after birth, she required intubation for severe dyspnea. Intravenous GCV was started leading to negative CMV DNA in blood on the eighth day of life with a residual viral load of 54,890 copies/mL in urine. Clinical conditions progressively deteriorated with severe respiratory and right heart failure refractory to maximizing ventilator support. After parental counseling, comfort care was started with exitus occurring on the 11th day of life.

Neonate n. 2 had a diagnosis of FGR at 27^2/7^ weeks. Maternal CMV serology was performed for the first time at 27^3/7^ weeks’ gestation, showing positive IgG (122 U/mL; CLIA, positive ≥ 22) and negative IgM (11.1 U/mL; CLIA, negative ≤ 12) with high IgG avidity (0.357; CLIA, high > 0.250). The woman declined amniocentesis. An emergency cesarean section was performed at 33^6/7^ weeks for worsening fetal conditions. The female neonate weighed 1050 g, with an HC of 28 cm. The Apgar score was 6 and 8 at the 1st and 5th minute, respectively. CMV DNA assessed on the first day of life was positive on blood and urine (Table 2). Blood exams revealed thrombocytopenia (53,000/μL), and cerebral MRI-diagnosed ventriculomegaly and polymicrogyria. ValGCV was started on the 10th day of life and was continued for 6 months. At the latest follow-up at 31 months of age, the infant shows cognitive delay and motor impairment, but no SNHL, with a physical growth around the 10th centile.

In case n. 3 there was a diagnosis of late-onset FGR at 36^1/7^ weeks, which led to the birth of a male neonate weighing 2300 g at 37^1/7^ weeks, with a regular HC. Positive CMV DNA testing was identified on the third day of life on urine (Table 2). All additional tests performed during the hospitalization were unremarkable. At the two-year follow up, the neonate showed a mild speech delay, which was confirmed at the subsequent visits.

Case n.6 had a diagnosis of FGR at 27^6/7^ weeks, which was confirmed at subsequent ultrasounds with an estimated fetal weight < 3rd centile. Maternal serology and virology for CMV were assessed for the first time at 35^4/7^ weeks with evidence of positive IgG and IgM and CMV DNA on urine sample (1350.0 copies/mL). Amniocentesis, performed at 36^5/7^ weeks, revealed presence of CMV DNA in the amniotic fluid (206,000,000 copies/mL) and the woman was started on VCV 8 g/day, according to the AIFA statement. Labor was induced at 38^1/7^ weeks for FGR, and a female neonate was born, with a weight of 1900 g. CMV DNA was positive on saliva and urine collected at birth (Table 2). Blood tests showed mild thrombocytopenia. Additional testing was unremarkable. A mild cognitive impairment and four-limb hypotonia were identified at the one-year follow-up.

## 4. Discussion

Here we report the results of the first three years of clinical use of an expanded targeted screening program for cCMV at our academic maternity center in Northern Italy.

Our findings are in line with previous studies investigating the clinical use of an expanded targeted cCMV screening program and identifying improved detection rates compared to a hearing-targeted screening program [14,26,27,28].

Hearing-targeted newborn screening programs have shown limitations regarding their yield of cCMV-infected neonates [6,14,23,37]. Only 25% of asymptomatic children show SNHL within the first month of life. Also, CMV-associated SNHL is uniquely characterized by fluctuating hearing levels, thus making the diagnosis of SNHL in the absence of a cCMV-dedicated follow-up even more challenging [5,7,8]. Notably, none of the eleven congenitally infected neonates identified in our three year-long study received cCMV testing because of a failed hearing screen.

Importantly, our incidence rate of 1.17% on all tested neonates is close to the 1.08% figure recently reported by Chiereghin and colleagues in a research study assessing a universal newborn cCMV screening program at their center located in Northern Italy [15]. Similar detection rates between expanded and universal cCMV screening programs have also been reported by studies conducted in the United State and Israel [14,27], further supporting the clinical relevance of expanded targeted screening.

In the absence of universal cCMV screening, the actual incidence of cCMV infection in the population is unknown, but the literature reports rates ranging from 0.2 to 1%, depending on maternal seroprevalence [2,38]. We identified a cCMV incidence rate of 0.14% among 7651 live-born infants, which is only one third or quarter of the expected incidence. Nonetheless, this figure is in line with published data, including the work by Zhang (0.14%) and Akiva (0.2%).

Four out of the eleven cCMV newborns identified in our study population were completely asymptomatic at birth with maternal infection as the sole indication for cCMV testing and one showed only deficient growth with low BW.

In a context where universal maternal serological screening for CMV is not recommended and newborn cCMV screening is performed only for failed hearing screenings, these five diagnoses of cCMV infection would have been missed.

During the study period, the available Italian guideline on uncomplicated pregnancies (2011) recommended against a universal CMV serological screening in pregnancy [32]. Notwithstanding this, Italian obstetricians have been used to prescribe such screening [39]. A suspected or confirmed maternal infection was only the sixth indication for cCMV testing (n = 42, 4.5%) but showed the highest prevalence of cCMV by identifying eight (8/42, 19%) cCMV-infected neonates, four of whom were asymptomatic. Of note, the Italian guideline has been recently updated (19 December 2023) and currently recommends universal CMV screening in the first and second trimester of pregnancy in women with negative serostatus [40]. Similarly, Canadian guidelines support universal CMV screening in pregnancy [41].

However, considering that CMV seroprevalence in women of childbearing age in Italy is approximately 65% [42] and that cCMV can occur not only after primary but also non-primary maternal infection [43], as observed in our cases n. 1, 5, and 8, inclusion of additional criteria in the expanded screening, such as BW < 10th centile, is extremely relevant. Importantly, BW < 10th centile was the main indication for cCMV testing in our study population, with a rate of 67.3% (n = 633), similar to that reported by Suarez and colleagues (68.2%) [14].

There were no cases with indication for cCMV infection screening who were not tested, and all exams were performed within 21 days of life, the temporal cut-off for differentiating congenital from postnatal infections [44]. These results suggest that the standardized medical record system we implemented was effective and avoided missed or delayed diagnoses.

The standard method for diagnosing cCMV infection is based on the identification of CMV DNA by PCR on a urine sample collected within 21 days of life [25]. Since collection of saliva is simpler than urine, saliva has been assessed as an alternative biological substrate for cCMV testing, reporting similar accuracy with negligible false positive rates (0.03–14%) due to breastfeeding [45,46,47].

Almost 96% of the neonates in our study population underwent screening by rt-PCR on saliva samples, with no cases of false positivity, further highlighting the feasibility and reliability of this screening method in the context of an expanded screening program.

Our study was conducted in an academic maternity center with a dedicated antenatal and postnatal clinic for perinatal infections, thus possibly limiting the generalizability of our findings. Also, our institution serves as a referral center for high-risk pregnancies identified in four nearby first level-care hospitals; therefore, our results may not be reflective of other centers given the potential referral bias. Inadequate antenatal care was included as a criterion for cCMV testing two years after the initial implementation of the expanded targeted cCMV screening at our institution, thus leading to a potential bias for modifying the inclusion criteria during the course of this study. Although with a retrospective design, our research findings are strengthened by the use of a standardized medical record system to identify all newborns with indications for cCMV testing, thus avoiding limitations related to patients’ chart reviews.

## 5. Conclusions

In conclusion, our data show that the implementation of an expanded targeted newborn screening program for cCMV is feasible and of clinical value, by improving the detection rate of congenital infection compared to targeted screening based only on failed hearing tests. Importantly, the observed rate of cCMV among tested neonates approximates that obtained in a similar geographical context by a universal screening program.

The recent implementation of a universal CMV serological screening program in pregnancy according to the 2023 updated Italian guideline on uncomplicated pregnancies would further strengthen the clinical relevance of such an expanded CMV screening program including maternal CMV infection as a testing criterion.

Further research on the feasibility and cost-effectiveness of an expanded targeted newborn screening program across different maternity and nursery settings is pivotal to inform clinical practice.

## Figures and Tables

**Figure 1 viruses-16-01138-f001:**
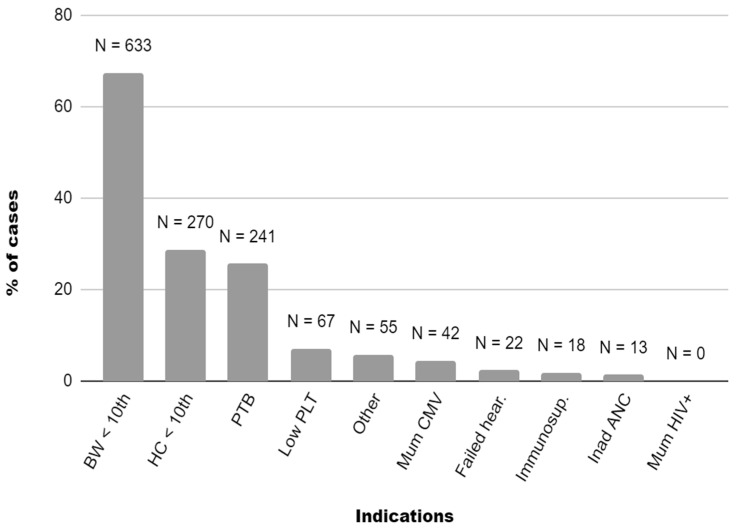
Indications for cCMV testing on 940 tested infants among 7651 live-born newborns followed up during a three-year program. Graphic representation of indications’ distribution for CMV testing. On the X axis there are indications for CMV testing in order of frequency. They are described as follows: birthweight < 10th centile (BW < 10th), head circumference < 10th centile (HC < 10th), preterm birth < 37 weeks’ gestation (PTB), thrombocytopenia (Low PLT), other neonatological indications (Other includes abnormal transcranial ultrasound findings and discrepancy between abdominal and head circumference), suspected or confirmed maternal CMV infection (Mum CMV), failed hearing screen (Failed hear.), maternal immunosuppression (Immunosup.), inadequate antenatal care (Inad ANC), and maternal HIV infection with detectable viral load and/or suppressed CD4+ count (Mum HIV+). Inadequate antenatal care was added as a criterion for neonatal testing in January 2023.

**Table 1 viruses-16-01138-t001:** General characteristics of the mothers and their newborns.

Maternal Characteristics (n = 904)	
Chronic diseases	169 (18.7)
Pregnancy-related pathological conditions	482 (53.4)
Multiple pregnancy	78 (8.6)
- Maternal CMV infection in pregnancyPrimary - Non-primary - Unknown	42 (4.6) 30 (3.3) 2 (0.2) 10 (1.1)
Vaginal birth	660 (70)
**Neonatal characteristics** **(n = 940)**	
Male gender	468 (49.8)
Term gestation (>37 wks)	699 (74.4)
Birthweight (grams)	2495.2 ± 600.7
Birthweight centile - <10th - >90th	17.2 ± 22.8 633 (67.3) 12 (1.3)
Head circumference centile - <10th - >90th	31.2 ± 26.6 270 (28.7) 32 (3.4)

The results are expressed as N (%) and mean ± SD. Chronic diseases include dysthyroidisms, pregestational diabetes, multiple sclerosis, maternal cardiopathy, chronic hypertension, epilepsy, and psychiatric disorders. Pregnancy-related pathological conditions include hypertensive disorders of pregnancy, gestational diabetes, dysthyroidisms, fetal growth restriction, fetal congenital malformation, placental anomalies, and preterm premature rupture of membranes.

**Table 2 viruses-16-01138-t002:** cCMV-infected newborns.

Patients	n. 1	n. 2	n. 3	n. 4	n. 5	n. 6	n. 7	n. 8	n. 9	n. 10	n. 11
Maternal CMV status at beginning of pregnancy	Seropositive	Unknown	Unknown	Seronegative	Seropositive	Unknown	Unknown	Seropositive	Seronegative	Seronegative	Seronegative
Indication for CMV testing	Mat. CMV NPI (peri-conception/1sttrim.) FGR (amnio+) BW < 10 PTB Low PLT Petechiae	FGRBW and HC < 10 PTB Low PLT Petechiae Hepattis	FGRBW < 10	Mat. CMV PI (24–28 weeks)	Mat. CMV NPI (peri-conception/1st trim.)	Mat. CMV PI (unknown timing) FGR (amnio+) BW and HC < 10 Low PLT	FGRBW < 10 Low PLT Petechiae	Mat. CMV NPI (peri-conception/1st trim.) Fetal ascites (amnio+) PTB Ascites	Mat. CMV PI (1st trim.)	Mat. CMV PI (26–30 weeks) BW and HC <10	Mat. CMV PI (1st trim.)
VCV in pregnancy	Yes	No	No	No	No	Yes	No	Yes	Yes	No	Yes
Gender	Female	Female	Male	Male	Female	Female	Male	Male	Female	Male	Male
Delivery mode	CS	CS	CS	VB	VB	VB	CS	CS	VB	VB	VB
GA (weeks)	33 5/7	33 6/7	37 1/7	37 5/7	40 3/7	38 1/7	37 5/7	36 1/7	39 3/7	41 4/7	39 6/7
BW centile	4	0	4	45	12	0	2	92	61	4	67
HC centile	35	1	25	70	33	0	70	97	79	0	74
NBHS result	Pass	Pass	Pass	Pass	Pass	Pass	Pass	Pass	Pass	Pass	Pass
CMV-positive specimens	Saliva, urine	Urine	Urine	Saliva, urine	Saliva, urine	Saliva, urine	Saliva, urine	Saliva, urine	Saliva, urine	Urine	Saliva, urine
Saliva VL (copies/mL)	19,498	-	-	50,000,000	44,296,505	20,575,137	2,699,368	13,619,118	18,000,000	-	13,000,000
Urine VL (copies/mL)	386,791	177,000,000	22,581,862	3,947,803	5,100,000	212,000,000	5,650,371	88,000,000	4,700,260	309,977	95,000,000
Whole blood VL (copies/mL)	477	155,808	22,581,862	2289	Neg.	4161	13,899	975	4128	390	17,150
Additional CNS findings (TCUS and MRI)	Left periventricular cystic lesion, ventriculomegaly	Ventriculomegaly, polymicrogyria	None	None	None	None	None	None	None	None	None
GCV ValGCV	Yes	Yes	No	Yes	No	No	No	Yes	No	Yes	No
Follow-up (length in months)	Deceased on 11th day of life	Cognitive delay and motor impairment (31 mo)	Mild speech delay (36 mo)	Abnormal bilateral ABR threshold at 2 months, then regular (25 mo)	Regular (26 mo)	Mild cognitive delay and motor impairment (15 mo)	Regular (6 mo)	Regular (9 mo)	Regular (9 mo)	Abnormal left ear ABR threshold at 3 weeks, then regular (7 mo)	Regular (6 mo)

Mat., maternal; CMV, cytomegalovirus; NPI, non-primary infection; PI, primary infection; FGR, fetal growth restriction; amnio+, amniocentesis positive for CMV DNA; BW, birthweight; HC, head circumference; PTB, preterm birth; low PLT, platelet (thrombocytopenia); VCV, valacyclovir; CS, cesarean section; VB, vaginal birth; GA, gestational age; NBHS, newborn hearing screening; VL, viral load; CNS, central nervous system; TCUS, transcranial ultrasound; MRI, magnetic resonance imaging; GCV, gancyclovir; valGCV, valgancyclovir; mo, months.

## Data Availability

Data are available upon reasonable request to the corresponding author.

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
