# Peer review of "Useful Clinical Criteria for Identifying Neonates with Congenital Cytomegalovirus Infection at Birth in the Context of an Expanded Targeted Screening Program"

_viruses, 2024, doi:10.3390/v16071138_

Round 1
Reviewer 1 Report
Comments and Suggestions for Authors
Authors performed an expanded targeted cCMV screening that tested 940 (12.3%) of 7,651 live born infants during three years. Eleven neonates were diagnosed as congenitally infected, for a prevalence of 1.17% on tested neonates and of 0.14% on live born infants. Authors determined that implementation of an expanded cCMV screening program appears feasible and of high clinical value. However, the incidence (0.14%) of cCMV diagnosed among 7,651 live born infants was only one third or quarter of the expected prevalence. Authors have to clarify many uncertainties in the manuscript.
Line 97
Authors added a criterion of inadequate antenatal care in January 2023. The inclusion criteria cannot be changed in a prospective cohort study. The efficacy of the expanded targeted cCMV screening must be evaluated without the additional criterion, or can be evaluated for neonates born after January 2023
Line 110
Not all mothers were tested for serological status of CMV. In a prospective cohort study, neonates from mothers who were not serologically tested cannot be included in study analyses. The inclusion criteria cannot be changed during a study period. Otherwise, it is an observational study but not a cohort study.
Results and Discussion were too descriptive, and have to be concise and clear in the revised manuscript.
Reviewer 2 Report
Comments and Suggestions for Authors
This is a small prospective study, conducted in a clinical setting (at a hospital that also attend to high risk pregnancies), that adds information on the added value of 'expanded targeted testing' to identify congenital CMV (cCMV) in newborns (in the absence of a universal testing programme for congenital CMV) (as is indeed the case in the majority of settings). The evolution of targeted cCMV testing has in the past focused on testing for CMV in babies who failed the (universal) newborn hearing screens (NBHS). The limitations of this approach (testing for cCMV at point of failed NBHS are recognised and include the fact that most cCMV babies are asymptomatic but a proportion (10- 15 %) will develop later onset symptoms, especially hearing loss, and so, will be missed by a targeted testing fosued on hearign loss. The authors comment that their expanded targeted testing for cCMV identified cCMV babies at a rate similar to a universal testing programme reported from 3 Italian sites. No infant was tested as a result of failed NBHS.
The authors could proffer some explanatory notes or comments on
- the very high urine viral loads (VL) vs somewhat lower saliva viral loads vs the lowest values in blood [ e.g. to me, this signifies that urine is still the best / and excellent source of CMV for diagnostic testing. That saliva has somewhat lower VL and blood even less - and that viraemia is not generally high in these congenitally infected newborns.
- small for gestational age (= SGA; small babies) is a common reason for testing babies for cCMV and a theme also in other studies. Can the authors comment on this in its utility for use as 'targeted' testing for cCMV? This would need a synthesis of other papers investigating SGA as one of the clinical triggers for testing a newborn for cCMV, but would be informative / provide some guidance for clinicians on the value of this approach. What would the rates of cCMV identification be in this study cohort if only SGA was used as a trigger for cCMV testing? I raise this as it is a common presenting and easily noticed feature in newborns if there is SGA (in fact- can be objectively documented when plotted on a growth chart) and thus an easily identified 'target' (as opposed to, say, hearing loss or low platelet counts).
Overall, well written and discussed report, that adds to the growing literature on expanded targeted testing programmes for cCMV.
Round 2
Reviewer 1 Report
Comments and Suggestions for Authors
1) This study is not a cohort study. Therefore, a term of “cohort” should be deleted throughout in the text.
2) The incidence (0.14%) of cCMV diagnosed among 7,651 live born infants was only one third or quarter of the expected prevalence. Authors have to explain this low efficacy for the detection of cCMV using their expanded cCMV screening program in Discussion.
3) Authors have to summarize what this study found newly and discuss clearly what is esprit of this study.
4) Discussion is still descriptive, and has to be concise and clear.
5) The inclusion criteria were changed during study periods. This is a weak point of this study should be excused in Discussion.
